# Study on Influencing Factors of Phase Transition Hysteresis in the Phase Change Energy Storage

**DOI:** 10.3390/ma15082775

**Published:** 2022-04-09

**Authors:** Dingyu Lu, Xiaofeng Xu, Xuelai Zhang, Wenhao Xie, Yintao Gao

**Affiliations:** Merchant Marine Collage, Shanghai Maritime University, Shanghai 201306, China; dylu@shmtu.edu.cn (D.L.); xlzhang@shmtu.edu.cn (X.Z.); xiewenhao1009@163.com (W.X.); fxx_xu@163.com (Y.G.)

**Keywords:** phase change energy storage, cryogenic phase change material, phase change hysteresis characteristic

## Abstract

Phase change energy storage is a new type of energy storage technology that can improve energy utilization and achieve high efficiency and energy savings. Phase change hysteresis affects the utilization effect of phase change energy storage, and the influencing factors are unknown. In this paper, a low-temperature eutectic phase change material, CaCl_2_·6H_2_O-MgCl_2_·6H_2_O, was selected as the research object, combined with the mechanism of phase change hysteresis characteristics, using a temperature acquisition instrument to draw the step cooling curve. A differential scanning calorimeter was used to measure the DSC (differential scanning calorimetry) curve, and the hysteresis characteristics of phase transformation were studied by factors, such as heat storage temperature, cooling temperature, and cooling rate. The experimental results show that when heating temperature increases by 30 °C, phase transition hysteresis decreases by about 3 °C. The cooling temperature decreased by 10 °C, and the phase transition hysteresis increased by 2.69 °C. This paper provides a new idea for optimizing the properties of phase change energy storage materials and provides a possibility for realizing the parametric control of phase change hysteresis factors.

## 1. Introduction

Now is the era of economic globalization, countries are trying to maximize the economy, the ensuing energy problem is more and more serious, energy shortage and environmental crisis is still a common topic in the world. Facing severe energy and environmental problems, it is more necessary to make efforts to save energy and reduce emissions. Among many energy-efficient utilization and recovery technologies, heat storage technology has attracted more and more attention, because there are certain regional and temporal differences between energy supply and demand. If there is a heat storage system between supply and demand for energy deployment, energy Internet can be realized smoothly [1]. Heat storage technologies mainly include sensible heat storage, latent heat storage and thermochemical reaction heat storage [2]. Compared with sensible heat storage and thermochemical reaction storage, latent heat storage with phase change material (PCM) as the main body has higher energy storage density and small temperature change, low-cost, and is easy to operate and control [3]. In the building energy-saving industry, electronic equipment temperature control, textile industry, automobile industry, solar energy, industrial waste heat utilization, and other fields have been more widely used [4,5,6,7,8,9]. As the key component of the energy storage unit, phase change energy storage material dominates the main performance of the whole energy storage unit. Optimizing the performance of phase change materials (PCMs) has become a core issue. The current international research on the phase change process is inadequate, it does not consider the phase change hysteresis phenomenon of energy storage materials, which has a significant impact on the charging and discharging performance of phase change energy storage materials. The hysteresis characteristics of phase change energy storage materials are as follows: the temperature range of phase change of energy storage materials is different in the process of heat storage and heat release, and there is a difference between melting temperature and crystallization temperature. Phase change hysteresis is a common phenomenon in the phase change process, which has a great influence on phase change materials. Therefore, it is necessary to systematically study the phase change hysteresis of phase change energy storage materials.

Phase change energy storage technology can improve energy utilization rate. As a key component of the energy storage unit, phase change materials can be divided into liquid–gas phase, solid–gas phase, solid–solid phase and solid–liquid phase according to phase change mechanism, but most of them adopt solid-to-liquid phase (melting) and liquid-to-solid phase (solidification) methods. As shown in Figure 1, the solid–liquid phase transition process is often considered to be an isothermal process, but this is only true for most pure substances. As shown in Figure 2, the measured solidification temperature of phase change materials is lower than the melting temperature, which is called phase change hysteresis [10,11]. In practice there is a temperature difference between melting and solidification, but in ideal there is no such difference. Therefore, phase transition hysteresis can be defined by the difference between the actual melting point and freezing point of phase change materials. This difference can also be used to measure the energy loss of a system [12]. The thermal hysteresis of many materials is due to the effects of subcooling or superheating, a phenomenon that needs to be carefully considered in PCM applications for renewable energy systems where temperature differences are small [13]. The hysteresis degree of phase change is too large so that the actual phase change temperature zone exceeds the operating temperature of the system, which affects both the endothermic and exothermic processes of phase change energy storage materials [14,15,16,17,18,19]. Existing studies have shown that, in the exothermic process of phase change materials, the phase change temperature in the solidification stage may be affected by the setting mode of materials, nucleating agent and other additives, thermal diffusivity, and latent heat value, which may be the main factors causing the phenomenon of undercooling and phase change hysteresis [20]. In order to understand these two factors affecting the phase transition process, scholars have analyzed the phase transition process in detail.

In recent years, many simulation studies in the field of phase change materials have adopted a simplified modeling method, that is, the melting and solidification processes are the same and described by enthalpy temperature curves. However, experimental results show that these processes are not identical, and the differences are caused by phase hysteresis. According to some reports, if the phenomenon is ignored, the significant difference between the simulation and the experimental results will be larger [21]. Ahmad et al. [22] of microcapsule phase change material wall panel established two kinds of numerical models, one is a model considering the material properties of solid–liquid interface, the other is a model of the apparent heat capacity method, after 400 thermal cycles, the experimental data and the data obtained from the two simulations were compared and analyzed, found that the apparent heat capacity method obtained result is closer to the experimental result. Enthalpy method only considers the initial and final states of PCM materials, while the apparent heat capacity method considers solid–liquid two-phase changes in the phase transformation process and takes the hysteresis of phase transformation into account, which is closer to the actual solid–liquid two-phase changes [23]. Therefore, the apparent heat capacity method is more suitable for the modeling and analysis of phase change materials. Buttitta et al. [24] found a large difference between melting point and freezing point on H (T) curve when conducting TEM experiments on microencapsulated phase change materials. The author attributed the difference between melting temperature and solidification temperature to phase transition hysteresis. Kuznik et al. [25] used DSC to conduct thermal analysis of paraffin-based microcapsule phase change materials and studied the melting and solidification process by using effective heat capacity for primary melting, primary solidification and continuous melting/solidification. It is found that the equivalent heat capacity of the melting (or solidification) process cannot accurately predict the heat capacity of the melting (or solidification) process. In addition, the equivalent heat capacity cannot accurately predict the temperature distribution of the continuous melting (or solidification) process. Therefore, phase transition hysteresis should be strictly considered in the numerical simulation of phase transition. In recent years, some scholars have proposed parameterizing the influencing factors to control the hysteresis phenomenon, so as to control the phase change process and weaken its influence on the energy storage characteristics of phase change materials. Khoury et al. [26] investigated the effect of thermal hysteresis phenomenon on the design parameters of TES systems. The accuracy of the simulation results with and without thermal hysteresis was investigated, and the effect of changes in design parameters and capsule envelope material in the presence of thermal hysteresis in spherical PCM capsules was assessed. They consider that PCM with strong thermal hysteresis will make the accuracy of melting time and usage temperature low. Efraín et al. [27] conducted a numerical study of the combined effects of phase change hysteresis in building envelope PCM walls. Developed a numerical code for simulates hysteresis, which uses an effective specific heat method to simulate the heat transfer through the PCM. Hysteresis was found to improve the thermal performance of PCM walls with a higher hysteresis temperature difference, but there is a limit to the improvement in thermal performance. Martin Zálešák et al. [28] developed a procedure to identify the effective heat capacity-temperature dependence and tested it on an air–PCM heat exchanger as a heat transfer problem with phase change. The relationship between effective heat capacity and temperature was found in the form of two independent ceff(T) curves to explain the phase change hysteresis. However, the accuracy of the results is somewhat low. There is a difference of 2.9% and 15.7% between the enthalpy of phase change obtained by differential scanning calorimetry (DSC) and the enthalpy of phase change obtained by solving the inverse problem. Delcroix et al. [29] calculated enthalpy-temperature curve and specific heat-temperature curve of PCM under different methods and different heat transfer rates by using inverse method, and found that phase transition hysteresis was related to heat transfer rate, and the higher the heat transfer rate, the greater the temperature difference of phase transition hysteresis. Hsu et al. [30] conducted experiments on three kinds of metal microcapsules PCMs (Zn/TiO_2_, Zn/Al_2_O_3_, Zn/SiO_2_), and the results showed that the phase change microcapsule materials had a large phase change lag, and the phase change lag degree increased with the increase in shell thickness and temperature gradient. Janabi et al. [31] studied the difference between the hysteretic model of phase transition and the enthalpy model, in which the difference varies with the intensity and time of solar radiation. Melting temperature, solidification temperature, latent heat of phase transition, solid material density, and material thickness are all important input parameters, that is, these parameters have great influence on phase transition hysteresis. Encapsulation can greatly change the thermal response of phase change materials (PCM) in terms of phase change temperature and thermal hysteresis. Karthikeyan Kumarasamy et al. [32] developed a novel numerical scheme based on computational fluid dynamics (CFD) to simulate the thermal response of EPCM and validated it with DSC curves of EPCM capsules.

The hysteresis of phase transition has a significant effect on the process of phase transition heat storage and release, so the research on the hysteresis of phase transition has a great prospect. At present, most of the studies are simulation and theoretical studies, without systematic sorting out the coupling relationship between the influencing factors and phase transition hysteresis. In this paper, a low-temperature eutectic composite phase change material CaCl_2_·6H_2_O-MgCl_2_·6H_2_O binary composite hydrated salt with phase change temperature of 27.5 °C and latent heat value of 173.2 kJ·kg^−1^ was selected. The composite phase change material has stable phase change temperature, high latent heat value of phase change, low undercooling, non-toxic, non-corrosive, and good economy. In this paper, the hysteresis characteristics of phase transition and its influencing factors in the process of heat storage and release are studied experimentally, which provides more ideas for the research direction of phase change hysteresis characteristics and plays an important role in promoting the development of phase change energy storage technology.

## 2. Experimental Materials and Equipment

The experimental material selected in this experiment is low-temperature eutectic composite phase change material CaCl_2_·6H_2_O-MgCl_2_·6H_2_O. The experimental instruments used in this experiment are shown in Table 1.

## 3. Study on Influencing Factors of Phase Transition Hysteresis

### 3.1. Effect of Heat Storage Temperature on Hysteresis Characteristics of Phase Transition

#### 3.1.1. The Experimental Operation

Taking CaCl_2_·6H_2_O-MgCl_2_·6H_2_O-2% SrCl_2_·6H_2_O-0.3% HEC as the research object, the heat storage and release experiments were carried out by changing the heat storage temperature, cooling temperature, cooling rate, the content of additive in the composite phase change material, and ultrasonic and other influencing factors. Phase change materials are independently developed by the laboratory. The phase transition point of the material is 25 °C, and the latent heat of phase transition is173.9 kJ·kg^−1^. Four groups of 40 g binary composite phase change materials CaCl_2_·6H_2_O-MgCl_2_·6H_2_O were weighed with an electronic balance and placed in four 50 mL small beakers, sealed with plastic film to prevent dehydration. The four groups of beakers were put into the thermostatic water tank with the temperature of 50 °C, 60 °C, 70 °C, and 80 °C, respectively, for water bath heating, and heated at a constant temperature until the material completely melted. During material melting, a temperature collector is used to collect temperature changes of phase change materials. After the material is fully melted, it is taken out and placed in a natural environment (the ambient temperature at this time is 10 °C) for cooling. When the temperature change of each sample during melting was measured by the temperature acquisition instrument, the sample temperature was recorded at an interval of 2 s, and the step cooling curve was drawn according to the relationship of temperature (T)/time (s). The temperature collection method of samples is shown in Figure 3. After the sample was taken out, the temperature of the phase change material finally reached the ambient temperature with the heat exchange between the sample and the surrounding environment. At this time, the data collection of the solidification process was completed. According to the relation of temperature (T)/time (s), the step cooling curve of the solidification process was obtained. The cooling temperature acquisition diagram is shown in Figure 4.

After the temperature collection of the melting process is completed, a small number of samples are taken out with a dropper and put into a paper beaker for the experiment of differential scanning calorimeter. There is a reference crucible (left) and sample crucible (right) in the test furnace, as shown in Figure 4a,b. The mass range of test samples was 5–10 mg. The DSC curve was obtained by setting the temperature program (linear heating, constant temperature for 3 min, cooling).

#### 3.1.2. Analysis of Experimental Results

##### Cooling Curve Analysis

As shown in Figure 5, the melting point and freezing point of phase change materials are different, and there is a gap between the two temperature points. The changing trend of the temperature difference of phase transition hysteresis with heat storage temperature is obtained.

As shown in Figure 6, with the increase in heat storage temperature, the temperature hysteresis of phase change materials gradually decreases, and the phase change hysteresis degree declines. The phase change hysteresis decreases from 4.25 °C at 50 °C to 1.52 °C at 80 °C. In addition, at high temperatures, such as 70 °C and 80 °C, it does not vary much in the degree of phase change hysteresis. The results show that the binary composite phase change material generally has phase change hysteresis characteristics, and the degree of phase change hysteresis is related to the heat storage temperature. Within the heating range allowed by constant temperature water bath, the higher the heat storage temperature, the lower the phase change hysteresis degree.

##### DSC Curve Analysis

Step cooling curve can be used to intuitively observe the temperature change with time during phase transformation of phase change materials. Step cooling curve can measure the temperature difference caused by the hysteresis characteristic of phase transition. However, other thermal properties of the low-temperature eutectic mixture cannot be obtained from the step cooling curve, and the temperature measured by the step cooling curve alone is not sufficient to support the analysis results, and experimental errors cannot be avoided. Therefore, DSC experimental analysis needs to be supplemented. The changing trend of phase transition temperature difference was verified, and the influence of heat storage temperature on phase transition hysteresis was further analyzed in combination with the enthalpy value of the eutectic mixture at low temperature. As shown in Figure 7, DSC characterization of phase transition properties of binary composite phase change materials under different heat storage temperatures. Table 2 shows the thermal performance parameters measured by DSC curves at different heat storage temperatures.

When using differential scanning calorimetry, the mass of the sample is generally between 5 and 10 mg, and it is not guaranteed that the obtained sample contains all kinds of added ingredients evenly in proportion. Therefore, the melting temperature and solidification temperature of binary composite phase change material characterized by DSC are not exactly the same as those measured by the step cooling curve, but the changing trend is the same, as shown in Figure 8.

After DSC measurement of the phase change material, it is found that the difference between the initial temperature point of melting phase change and the initial temperature point of solidification phase change of the phase change material decreases with the increase in heat storage temperature, which is consistent with the result of temperature (T)–time (S) curve measurement. Melting enthalpy and crystallization enthalpy did not change significantly at different temperatures. The phase change hysteresis decreases from 4.92 °C at 50 °C to 1.94 °C at 80 °C, which decreased by 2.98 °C. It shows that the increase in heat storage temperature will affect the melting temperature and solidification crystallization temperature of phase change materials, and then change the difference between melting temperature and solidification crystallization temperature, and form a certain rule. Combined with the step cooling test results and the performance characterization results of DSC, it can be concluded that the hysteresis degree of phase transition temperature is related to the heat storage temperature. Within the allowed heating range, the higher the heat storage temperature, the smaller the hysteresis degree of the phase transition temperature. An increase in heating temperature easily causes a large temperature difference, and the phase change material will quickly absorb heat, heat flow density is larger, and heat is uniform.

### 3.2. Effect of Cooling Temperature on Hysteresis Characteristics of Phase Transition

#### 3.2.1. The Experimental Operation

Six 40 g binary composite phase change materials (CaCl_2_·6H_2_O-MgCl_2_·6H_2_O) were weighed by an electronic balance and placed in a constant temperature water bath at 50 °C. Temperature changes of phase change materials during melting were measured by temperature acquisition instrument. When the phase change material is heated at a constant temperature until it is completely melted and the temperature is kept at 50 °C, the phase change material is placed in an alcohol constant temperature and low-temperature tank with a temperature of 2 °C, 4 °C, 6 °C, 8 °C, and 10 °C for cooling, and the temperature change of the sample is recorded with a temperature collector, as shown in Figure 9. For both melting and solidification processes, sample temperatures were recorded at 2 s intervals, and step cooling curves were drawn according to the relationship of temperature (T)/time (s).

#### 3.2.2. Analysis of Experimental Results

##### Cooling Curve Analysis

Six groups of samples with the same ratio were heated in a constant temperature water bath of 50 °C, and the measured heat storage curves showed little difference. The exothermic process is different, especially the crystallization temperature in the solidification stage. Figure 10 shows the melting initiation temperatures of samples at different cooling temperatures.

Table 3 shows the melting initiation temperature and crystallization initiation temperature measured at different cooling temperatures, and the temperature lag difference of phase transition temperature of each group of materials is plotted, as shown in Figure 11. With the decrease in cooling temperature, the crystallization temperature of the exothermic process will change, and the melting temperature of the heating process is independent of the cooling final temperature. The temperature difference between the initial solidification temperature and the initial melting temperature will vary with the cooling end temperature. The lower the cooling end temperature is set, the greater the temperature lag difference between the two initial phase transition temperatures and the greater the degree of phase transition lag. When the cooling temperature is 10 °C, the phase transition lag is the smallest, which is 1.44 °C. When the cooling temperature is 0 °C, the phase transition lag is the largest, which is 4.13 °C.

##### DSC Curve Analysis

Differential scanning calorimeter was used to test six groups of phase change materials, which were heated to 50 °C at a rate of 5 °C per minute and then cooled to 10 °C, 8 °C, 6 °C, 4 °C, 2 °C, and 0 °C, respectively (Cooling rate is 5 °C per minute). The change of heat flux with temperature was measured and DSC curves were obtained to analyze the melting temperature, solidification temperature, and latent heat value of phase change, as shown in Figure 12 and Table 4.

On the whole, the difference of phase transition temperature between the heat storage process and the heat release process shows an increasing trend as the cooling final temperature decreases, as shown in Figure 11. Therefore, there is a phase transition hysteresis between the phase transition temperature of heat storage and the heat release process, and the lower the cooling temperature is set, the greater the phase transition temperature hysteresis difference and the greater the phase transition hysteresis degree, which is similar to the results of step cooling test. It is worth noting that the phase transition hysteresis of DSC test results is larger than that of the step cooling test. Because the sample size of dsc measurement is small, the cooling method is liquid nitrogen air cooling, the response of temperature control is slower, and the uniformity of heating is poor. The step cooling curve thermostatic bath sample volume is large, the cooling method is anhydrous ethanol liquid cooling, the heat transfer is relatively uniform, resulting in melting crystallization phase change lag is smaller.

### 3.3. Effect of Cooling Rate on Hysteresis Characteristics of Phase Transition

#### 3.3.1. The Experimental Operation

Four 40 g binary composite phase change materials (CaCl_2_·6H_2_O-MgCl_2_·6H_2_O) were weighed with an electronic balance, and placed in a constant temperature water bath at 50 °C, and heated at a constant temperature until the material completely melted. During the melting process, a temperature collector was used to collect the temperature change of the phase change material. After the material fully melted, the samples were taken out. The fan was set to four different wind speeds, and the wind speed was measured by an anemometer. The wind speed was set to 2.5 m/s, 3.5 m/s, 4.5 m/s, and 5.5 m/s. The fan was fixed in the same position to blow the sample directly, and a temperature acquisition instrument was used to record the temperature change of the sample in the heat release process, as shown in Figure 13a,b.

#### 3.3.2. Analysis of Experimental Results

##### Cooling Curve Analysis

Temperature changes were recorded at 2 s intervals in both melting and solidification processes, and step cooling curves were drawn according to the relationship between temperature and time. The temperature (T)/time (s) curves of the four groups of samples were heated to 50 °C, and the various rules and trends of the temperature (T)/time (S) curves of the heating section were roughly similar. Therefore, only the initial temperature of phase transition in the heat storage process was counted in Table 5. The temperature-time variation curves of the four groups of samples in the heat release process are summarized in Figure 14.

The hysteresis difference of phase change temperature of the binary composite phase change material was drawn as the curve in Figure 15, and its changing trend with the cooling rate was observed. With the decrease in the cooling rate, the crystallization temperature will change during the exothermic process, and the melting temperature during the heating process is independent of the cooling rate. The temperature difference between crystallization initiation temperature and melting initiation temperature will vary with the cooling rate. When the wind speed is 2.5 m/s, the phase transition lag is the largest, which is 6.26 °C. When the wind speed is 5.5 m/s, the phase transition lag is the smallest, which is 0.49 °C. The lower the cooling rate is set, the greater the temperature lag difference between the two initial phase transition temperatures and the greater the degree of phase transition lag.

##### DSC Curve Analysis

Four groups of phase change materials were tested by differential scanning calorimeter. Heating at a rate of 5 °C per minute until complete melted. The samples were heated to 50 °C, and the cooling rate was changed during the cooling process. The cooling rate was set to 5 °C/min, 6 °C/min, 7 °C/min, 8 °C/min, 9 °C/min, and 10 °C/min, respectively. The change of heat flux with temperature was measured, and the DSC curve was obtained, as shown in Figure 16. The melting temperature, crystallization temperature, and latent heat of phase transformation were calculated in Table 6.

According to the results of melting initiation temperature, crystallization initiation temperature, and temperature difference of binary composite phase change materials, the temperature difference of phase change hysteresis gradually decreases with the increase in cooling rate, as shown in Figure 17. This trend is the same as the results of step cooling curve test. The overall trend of the hysteresis of phase transition temperature caused by the cooling rate is that the temperature hysteresis gradually decreases with the increased cooling rate. When the cooling rate is 5 °C/min, the phase transition lag reaches a maximum of 8.86 °C. When the cooling rate is 10 °C/min, the phase transition lag reaches a minimum of 5.12 °C, which decreased by 3.74 °C. It can be seen that there is phase transition hysteresis between the phase transition temperatures in the heat storage and heat release processes, and the higher the cooling rate, the greater the temperature hysteresis difference of the material, so the greater the phase transition hysteresis degree. The higher the cooling rate is, the more uneven the PCM is heated, so the crystallization and melting are not complete, which will lead to the increase in phase transition hysteresis. Slow heating and cooling of phase change materials can better melt and solidify.

## 4. Conclusions

In this paper, a low-temperature eutectic composite phase change material, CaCl_2_·6H_2_O-MgCl_2_·6H_2_O, phase change temperature of 27.5 °C, the latent heat value of 173.2 kJ·kg^−1^, was selected. Taking the phase change material as the research object, the experimental research on the influencing factors of phase change hysteresis is introduced in detail, including the experimental ideas and detailed experimental steps, and the following conclusions are obtained from the analysis:(1)Heat storage temperature is an influential factor affecting the hysteresis of phase transition of energy storage materials, and the hysteresis degree of phase transition of energy storage materials will change with the change of heat storage temperature. Through the step cooling experiment and differential scanning calorimeter test, the same results are obtained: within the allowed heating range, the higher the heat storage temperature is set, the smaller the difference of phase transition hysteresis temperature of the energy storage material is, and the smaller the phase transition hysteresis degree of the energy storage material is;(2)The final cooling temperature is a factor that affects the hysteresis of phase transformation of energy storage materials. The hysteresis of phase transformation of energy storage materials will be changed by changing the terminal temperature of the cooling process. The step cooling curve obtained by the temperature collector is consistent with the DSC curve obtained by the differential scanning calorimeter: the lower the cooling end temperature is set, the greater the temperature lag difference of the energy storage material is, and the greater the phase transition lag degree of the energy storage material is. The cooling rate during the cooling process is an important factor affecting the hysteresis of phase transformation of energy storage materials. Cooling curves which are obtained by temperature acquisition instrument, and obtained by differential scanning calorimeter DSC curve of the conclusion, the conclusion is the same: as the cooling rate drop, the temperature difference of energy storage materials varies with the cooling rate, set a lower cooling rate, the temperature of the energy storage material lags behind the difference, the greater the higher the degree of transformation hysteresis;(3)The homogeneous heat transfer of phase change materials greatly affects the hysteresis of phase change. The slow heating and cooling of phase change materials can complete the phase change process well and avoid the phenomenon of phase change hysteresis. Factors affecting the heat transfer process of phase change materials will affect the phenomena after phase change, which is also a future research direction.

## Figures and Tables

**Figure 1 materials-15-02775-f001:**
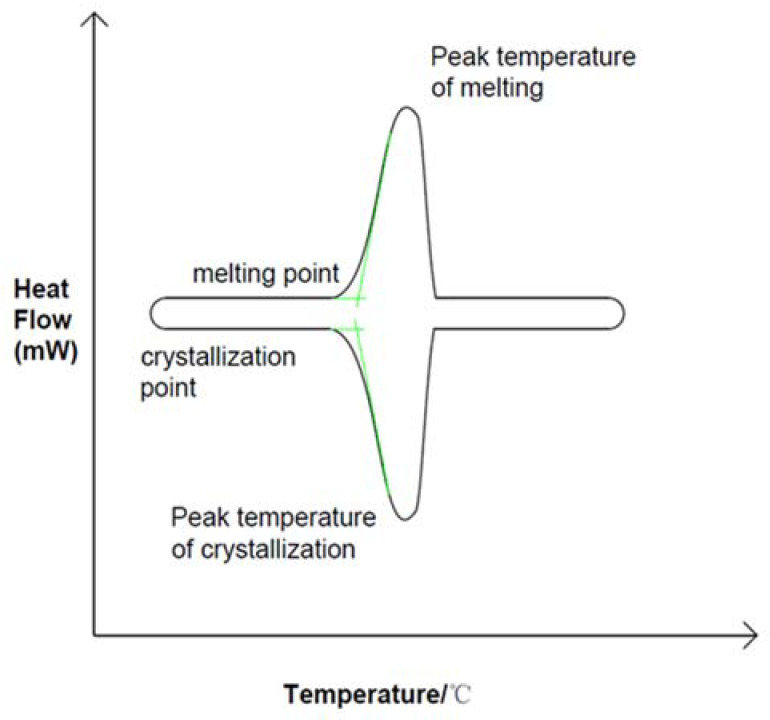
DSC curve of ideal phase transition, Reprinted with permission from ref. [10]. Copyright 2020 Elsevier.

**Figure 2 materials-15-02775-f002:**
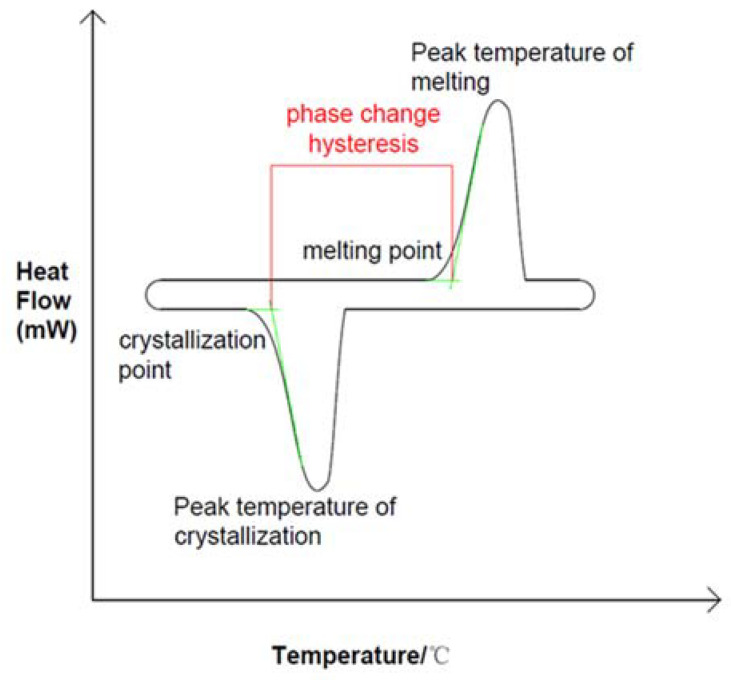
Phase transition hysteresis in DSC curve during actual phase transition, Reprinted with permission from ref. [10]. Copyright 2020 Elsevier.

**Figure 3 materials-15-02775-f003:**
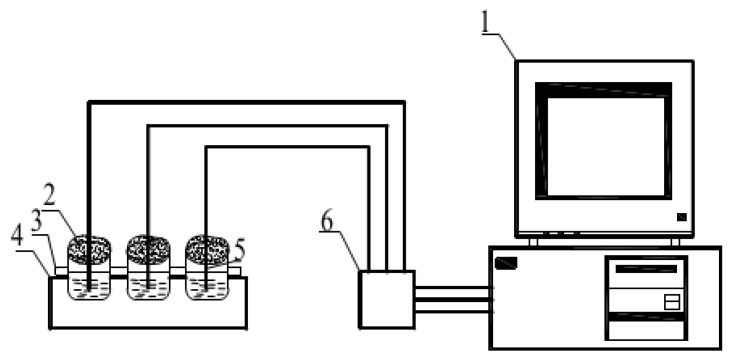
Schematic diagram of step cooling experimental system. 1—Computer; 2—Beaker; 3—Thermal insulation layer; 4—Cryogenic tank; 5—Thermocouple; 6—Data collector

**Figure 4 materials-15-02775-f004:**
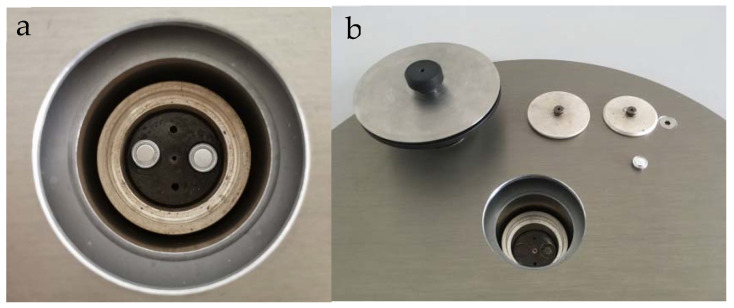
(**a**) Physical drawing of DSC test furnace cavity. (**b**) Reference sample (left) Test sample (right).

**Figure 5 materials-15-02775-f005:**
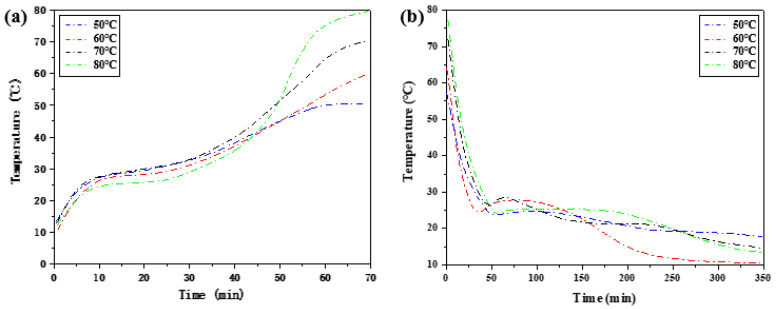
Heat storage and release step cooling curves of composite phase change materials at different heat storage temperatures. (**a**) Step heating curve of the heat storage process. (**b**) Step cooling curve of the exothermic process.

**Figure 6 materials-15-02775-f006:**
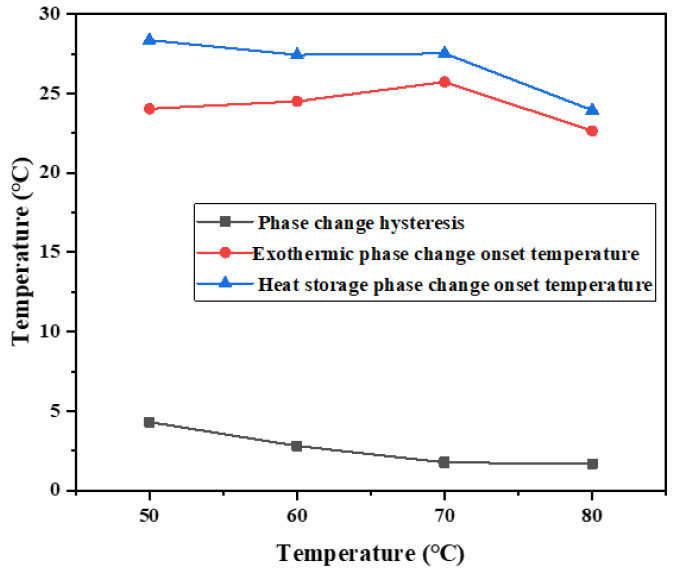
Influence of different heat storage temperatures on hysteresis characteristics of phase transition.

**Figure 7 materials-15-02775-f007:**
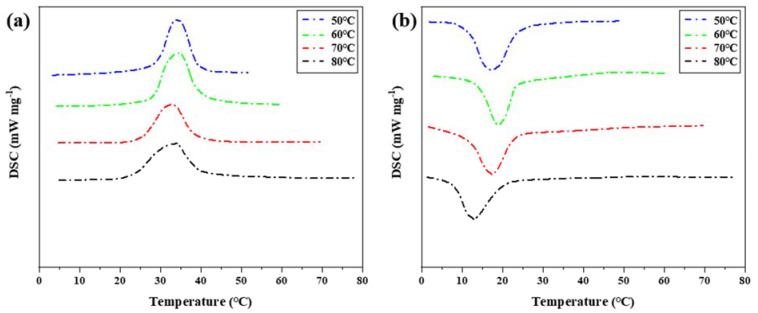
DSC characterization of phase transition properties of binary composite phase change materials at different heat storage temperatures (**a**) DSC curve of the second temperature rise. (**b**) DSC curve of the first cooling.

**Figure 8 materials-15-02775-f008:**
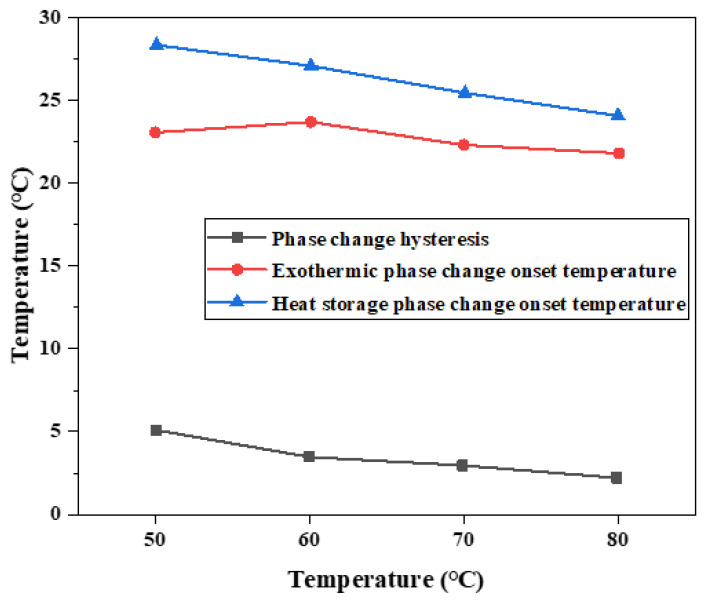
Influence of different heat storage temperatures on hysteresis characteristics of phase transition.

**Figure 9 materials-15-02775-f009:**
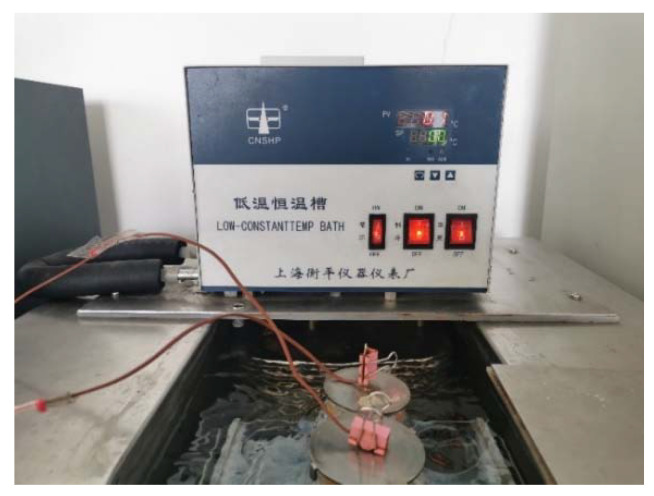
Actual sample of constant temperature alcohol bath cooling.

**Figure 10 materials-15-02775-f010:**
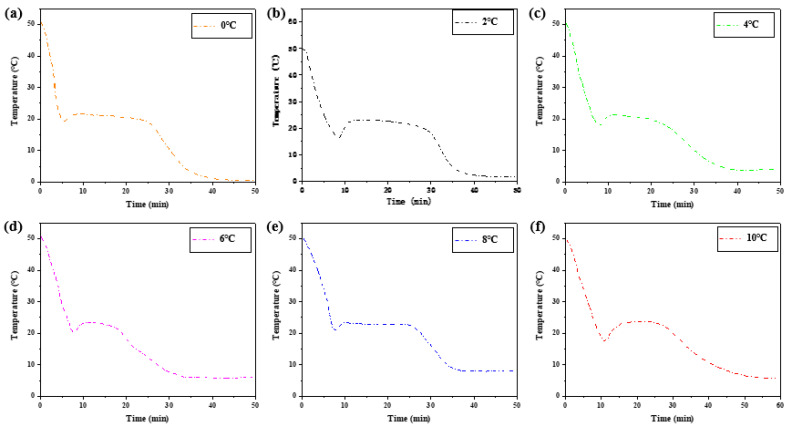
The exothermic process at different cooling temperatures.

**Figure 11 materials-15-02775-f011:**
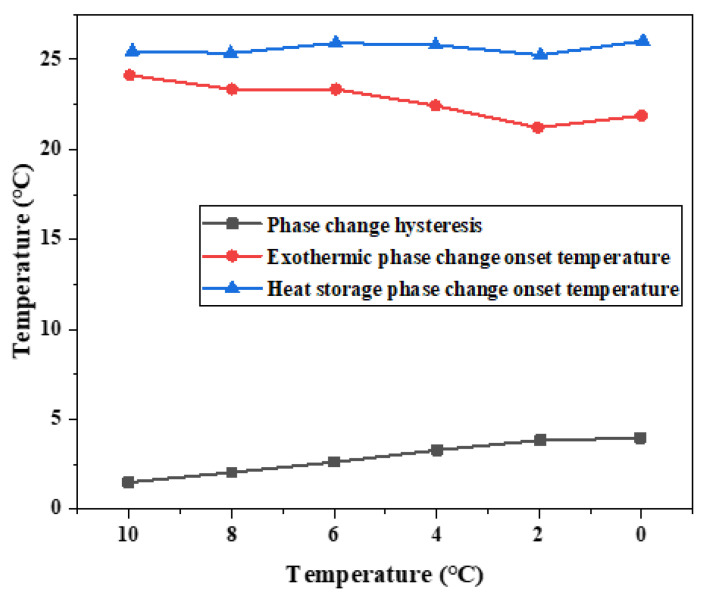
Influence of different cooling temperatures on hysteresis characteristics of phase transition temperature.

**Figure 12 materials-15-02775-f012:**
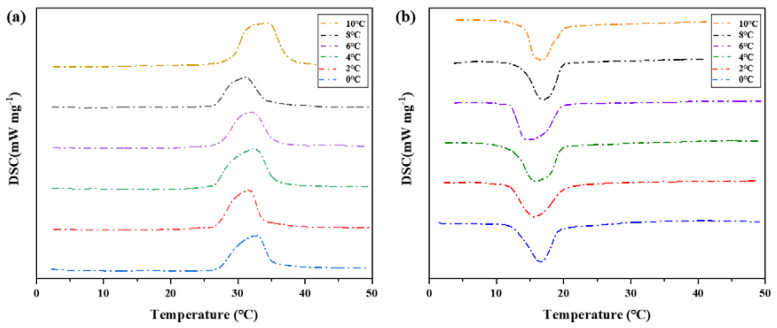
DSC characterization of phase transition properties of binary composite phase change materials at different cooling temperatures. (**a**) DSC curve of the second temperature rise. (**b**) Step cooling curve of the exothermic process.

**Figure 13 materials-15-02775-f013:**
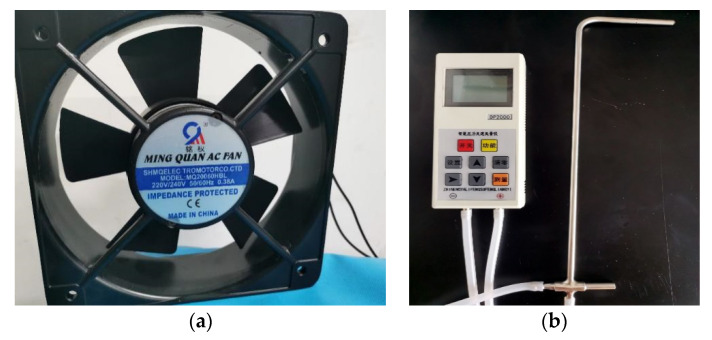
(**a**) Physical drawing of adjustable speed experimental fan. (**b**) Anemometer.

**Figure 14 materials-15-02775-f014:**
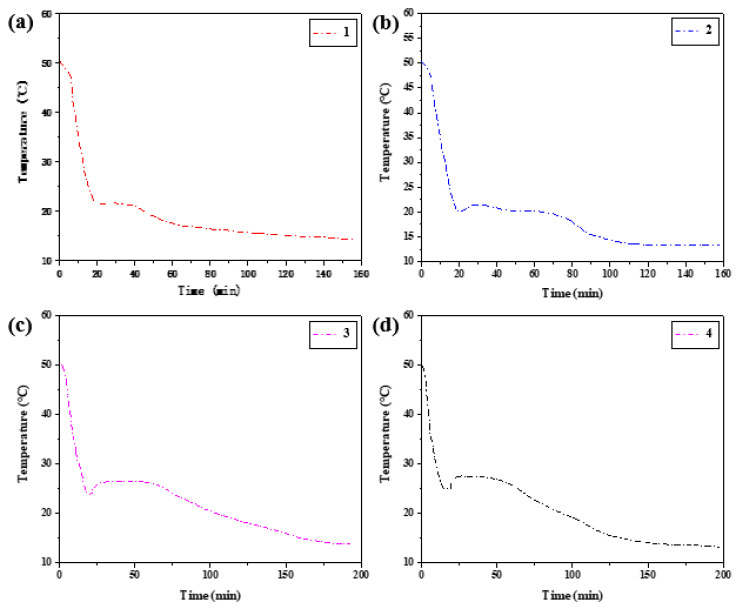
Step cooling curves of exothermic process at different cooling rates.

**Figure 15 materials-15-02775-f015:**
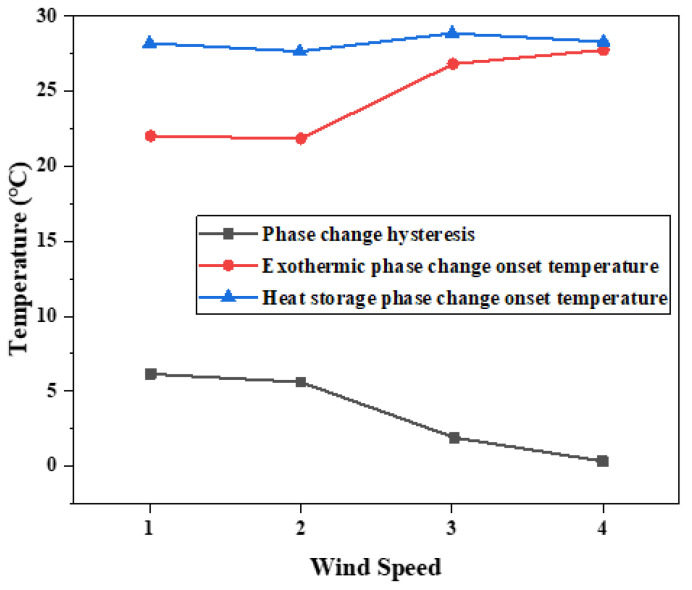
Influence of different cooling rates on hysteresis characteristics of phase transition.

**Figure 16 materials-15-02775-f016:**
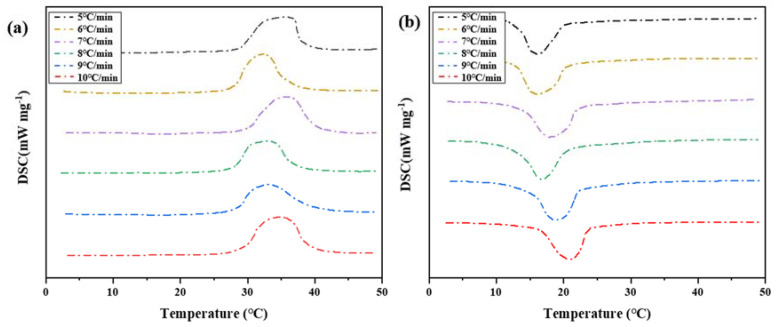
DSC characterization of phase transition properties of binary composite phase change materials at different cooling rates. (**a**) DSC curve of the second temperature rise. (**b**) DSC curve of the first cooling.

**Figure 17 materials-15-02775-f017:**
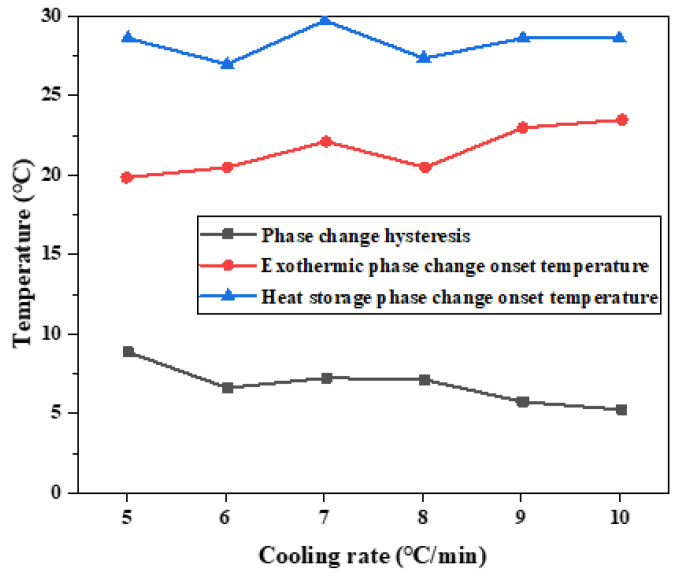
Influence of different cooling rates on hysteresis characteristics of phase transition.

**Table 1 materials-15-02775-t001:** Table of experimental instruments.

Equipment	Model	Precision
Precision electronic balance	MSl05DU	±0.01 mg
Electronic balance	FA2004	±0.1 mg
Differential scanning calorimeter	DSC200F3	<0.1 °C
Agilent data acquisition instrument	34972A	±0.01 °C
High and low temperature thermostatic water tank	DC-6515	±0.01 °C
Alternating high and low temperature impact chamber	YSGJW-100C	±0.5 °C
Scanning electron microscope	KYKY-EM6000	—
Low temperature drying oven	DECD-80	±0.1 °C
Magnetic stirrer	90–2	—
T-thermocouple	—	±0.05 °C
Thermal conductivity tester	TPS 2500s	±3%
Thermocouple spot welder	—	—
Beaker, dropper, test tube, etc.	—	—
Fourier infrared spectrometer	TENSOR37	8000–350
X-ray diffractometer	DX-2700	≤0.001°
Electric blast drying oven	DHG-9140A	±0.1 °C
Ultrasonic breaker	WM-1000	

**Table 2 materials-15-02775-t002:** Thermal performance parameters measured by DSC curves at different heat storage temperatures.

Serial Number	Melting Process		Crystallization Process		
	Initial Temperature (°C)	Melting Enthalpy (J/g)	Initial Temperature of Solidification (°C)	Crystallization Enthalpy (J/g)	Temperature Difference (°C)
1 (Heating to 50 °C)	28.01	173.40	23.08	145.20	4.92
2 (Heating to 60 °C)	27.50	169.30	23.86	144.10	3.64
3 (Heating to 70 °C)	25.60	173.50	22.47	128.60	3.13
4 (Heating to 80 °C)	23.80	175.20	21.86	136.90	1.94

**Table 3 materials-15-02775-t003:** Melting initiation temperature and crystallization initiation temperature measured at different cooling temperatures.

Different Cooling Temperature (°C)	10	8	6	4	2	0
Initial temperature of melting phase transition (°C)	25.43	25.44	25.97	25.85	25.07	26.04
Initial temperature of solidification phase transition (°C)	23.99	23.25	23.29	22.48	21.21	21.91
Temperature lag difference (°C)	1.44	2.20	2.68	3.37	3.86	4.13

**Table 4 materials-15-02775-t004:** Thermal performance parameters measured by DSC at different cooling temperatures.

Serial Number	Melting Process		Crystallization Process		Temperature Difference (°C)
	Initial Temperature (°C)	Melting Enthalpy (J/g)	Initial Temperature of Solidification (°C)	Crystallization Enthalpy (J/g)	
1 (Cooling to 10 °C)	26.50	151.90	21.60	158.20	4.90
2 (Cooling to 8 °C)	26.20	172.60	20.10	167.10	6.10
3 (Cooling to 6 °C)	26.30	164.80	19.50	140.20	6.80
4 (Cooling to 4 °C)	26.40	154.60	19.60	131.70	6.80
5 (Cooling to 2 °C)	26.90	167.40	20.60	146.30	6.30
6 (Cooling to 0 °C)	27.30	170.80	20.10	153.40	7.20

**Table 5 materials-15-02775-t005:** Melt initiation temperature and crystallization initiation temperature measured under different wind speed conditions.

Different Wind Speed	1/2.5 m/s	2/3.5 m/s	3/4.5 m/s	4/5.5 m/s
Initial melting temperature (°C)	28.30	27.80	28.90	28.30
Crystallization initiation temperature (°C)	22.04	22.03	26.85	27.81
Temperature lag difference (°C)	6.26	5.77	1.45	0.49

**Table 6 materials-15-02775-t006:** Thermal performance parameters measured by DSC at different cooling rates.

Serial Number	Melting Process		Crystallization Process		
	Initial Temperature (°C)	Melting Enthalpy (J/g)	Initial Temperature of Solidification (°C)	Crystallization Enthalpy (J/g)	Temperature Difference (°C)
1 (5 °C/min)	28.70	116.80	19.84	102.90	8.86
2 (6 °C/min)	27.00	126.00	20.40	104.10	6.59
3 (7 °C/min)	29.80	143.40	22.14	104.60	7.26
4 (8 °C/min)	27.40	131.10	20.41	106.20	6.99
5 (9 °C/min)	28.50	142.10	22.91	107.90	5.59
6 (10 °C/min)	28.57	172.40	23.45	143.00	5.12

## Data Availability

Not applicable.

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
