# Peer review of "Study on Influencing Factors of Phase Transition Hysteresis in the Phase Change Energy Storage"

_materials, 2022, doi:10.3390/ma15082775_

Round 1

Reviewer 1 Report

In this paper, the authors proposed an experimental study on hysteresis characteristics of phase change material investigating the factors such as heat storage temperature, cooling temperature, cooling rate.

In particular, the study is based on a low-temperature eutectic phase change material, CaCl2 6H2O-11 MgCl2 6H2O, using a temperature acquisition instrument to draw the step cooling curve.

In general, this topic falls within the aim and scope of the Materials journal.

The scientific context is interesting, but in my opinion some aspects need to be clarified:

  • Abstract: Should have one sentence for each: context and background, motivation and novelty, hypothesis, methods, results with values, conclusions.
  • Introduction: The Introduction section could be thoroughly improved. A robust comparison with other approaches and results published in this field in high impact journals should be presented. Also, I suggest the authors to consider the following paper (https://doi.org/10.1016/j.enconman.2019.112101) between lines 37-38.
  • Introduction from line 124 to 137: Why this paper is presented? The authors should define the motivations, the innovative aspects respect to the other studies already published. I advise the authors to better specify the innovative contribution with respect to the papers on hysteresis already published and present in the literature. What is the added value? The innovation of the aforementioned work compared to what has already been published?
  • Figures 1 and 2 are not made by the authors but are taken from this article (https://www.sciencedirect.com/science/article/pii/S0167732220311934?via%3Dihub), therefore it is necessary to quote the paper in the captions and ask copyright permission to the publisher.
  • Table 2, 3, 4, 5, 6 and 7 the celsius degree symbol is staggered. please correct it
  • In figure 12 b, the specimen starts from a T equal to 55 ° C, could this compromise the final result?
  • Line 300 authors sated “as shown in Figure 15”. I think the reference to the figure 15 is wrong, please check.
  • From line 304 to 306, the results obtained with DSC are much higher than the experiment, justify the reason.
  • In figure 16 a, the specimen starts from a T equal to 55 ° C and not 50 ° C, could this compromise the final result?
  • From line 353 to 354. Do these rates ensure the same cooling obtained with the experiment of 2.5m/s, 3.5m/s, 4.5m/s and 5.5m/s?

In conclusion, the document as it presents itself shows some unclear aspects, but it could turn out to be a good job if carefully revised. Therefore, the authors should carry out a review process in order to significantly improve the article.

Author Response

Dear editors and reviewers:

Thank you very much for reviewing my paper. I have revised the paper according to your opinions and marked it in red. The following are the modification instructions.

  • Abstract: Should have one sentence for each: context and background, motivation and novelty, hypothesis, methods, results with values, conclusions.

I have modified it.

  • Introduction: The Introduction section could be thoroughly improved. A robust comparison with other approaches and results published in this field in high impact journals should be presented. Also, I suggest the authors to consider the following paper (https://doi.org/10.1016/j.enconman.2019.112101) between lines 37-38.

I have cited the literature.

  • Introduction from line 124 to 137: Why this paper is presented? The authors should define the motivations, the innovative aspects respect to the other studies already published. I advise the authors to better specify the innovative contribution with respect to the papers on hysteresis already published and present in the literature. What is the added value? The innovation of the aforementioned work compared to what has already been published?

At present, most of the researches are simulation and theoretical researches, without systematic sorting out the coupling relationship between the influencing factors and phase transition hysteresis. In this paper, the effect of solidification - melting temperature on the phenomenon is studied experimentally in the hope of controlling the hysteresis of phase transition.

  • Figures 1 and 2 are not made by the authors but are taken from this article (https://www.sciencedirect.com/science/article/pii/S0167732220311934?via%3Dihub), therefore it is necessary to quote the paper in the captions and ask copyright permission to the publisher.

I have marked the reference, and the reference is published by our research group.

  • Table 2, 3, 4, 5, 6 and 7 the celsius degree symbol is staggered. please correct it

I have modified it.

  • In figure 12 b, the specimen starts from a T equal to 55 ° C, could this compromise the final result?

I have modified it.

  • Line 300 authors sated “as shown in Figure 15”. I think the reference to the figure 15 is wrong, please check.

I have modified it.

  • From line 304 to 306, the results obtained with DSC are much higher than the experiment, justify the reason.

Because the sample size of dsc measurement is small, the cooling method is liquid nitrogen air cooling, the response of temperature control is slower and the uniformity of heating is poor. Step cooling curve thermostatic cold bath sample volume is large, the cooling method is anhydrous ethanol liquid cooling, the heat transfer is relatively uniform, resulting in melting crystallization phase change lag is smaller.

  • In figure 16 a, the specimen starts from a T equal to 55 ° C and not 50 ° C, could this compromise the final result?

I have modified it.

  • From line 353 to 354. Do these rates ensure the same cooling obtained with the experiment of 2.5m/s, 3.5m/s, 4.5m/s and 5.5m/s?

We agree from the complete phase transition and temperature

Reviewer 2 Report

The paper reports on the influencing factors of phase transition hysteresis in phase change materials (PCM). An experimental investigation of heat charging and discharging has been conducted to detect these factors. The elucidation of the origin of phase transition hysteresis is important for the scientific and engineering community who work with PCMs in energy storage applications. Although the authors presented some influencing factors, the materials and methods used are not well established and elaborated. Therefore, these suggestions are given to improve the present manuscript's quality.

  1. The English language is moderate and needs more improvement. Especially, long sentences should be avoided as their meaning get lost along the way. For example, lines 41-46: “At present, the research…phase change energy storage materials.” Lines 53-56, lines 85-90 ,and much more.
  2. Figures 1 and 2 can be combined to make a comparison between the ideal and actual phase change material behavior.
  3. In introduction line 95, Therefore …“hot melt method”… It appears that this method has not been mentioned prior this sentence, and should be elaborated.
  4. What is “reverse thinking method” in line 112? It would be best if you used the proper terminology.
  5. The sentences in lines 41-45 and lines 124-127 are similar, almost redundant. Please, consider revising.
  6. In the “materials and methods section”, what is the composition of your material. Is it equimolar or equi-mass mixture of salt hydrates MgCl2.6H2O -CaCl2.6H2O? The detail on the preparation of the mixture is unclear.
  7. Figure 3 caption is not correct? Isn’t step heating process? Also, consider combining Figure 3 and 4 as they portray the same idea: step heating and cooling process.
  8. Likewise, combine Figures 5 and 6 of the DSC.
  9. Check the legend of Figure 7(a), I think there is a mixup of green and black dash lines. Don’t you think? Check also the caption of Figure 7(a). step cooling?
  10. Table 2 and Figure 8 present the effects of heat charging temperature on the phase transition hysteresis. Consider using only one to avoid redundancy. Similarly, consider revising throughout the manuscript, Table 3 and Figure 10, and so on.
  11. What is the heating rate in the DSC analysis of Figure 9?
  12. It is very difficult to understand the experimental procedure in section 3.2.1, especially lines 251-255. In fact, the experimental procedure is uncoherent: in section 3.1.2 the PCM is heated in a water bath and then cooled in natural convection, while in section 3.2.1 the PCM is heated in water and cooled in alcohol? Why are you choosing two different methods of cooling ? Those are some concerns that make this study unreliable.
  13. Figure 11 caption is inconsistent with comments in lines 251-255. Is it cooling in water or alcohol?
  14. In Table 4, where do you get the melting point values, as the heating step was not mentioned in the experimental section.
  15. The cooling and heating rate of the DSC in Figure 14 are not given.
  16. Why did you choose different starting weight of materials, 50 mg in section 3.1 and 40 mg in sections 3.2 and 3.3?
  17. Figure 15 and 16 can be put together.
  18. In lines 351-356, it is said that the samples were heated to 50 o what was the heating rate? Was it variable as shown in Figure 18 (a)?

In summary, this work was a little bit too hard to read and digest. The oresentation of the manuscript is very poor and should be completely revised. The author should thoroughly describe the experimental section and present the results separately. Therefore, sections 3.1.1, 3.2.1 and 3.3.1 should be moved to the experimental sections. Furthermore, the main factors of phase change hysteresis were not given. The authors only discussed the exogenous factors such as heat storage temperature, cooling storage temperature. However, how about the endogenous factors? The authors should elaborate on that or give some hints for future directions .

Author Response

Dear editors and reviewers:

Thank you very much for reviewing my paper. I have revised the paper according to your opinions and marked it in red. The following are the modification instructions.

The English language is moderate and needs more improvement. Especially, long sentences should be avoided as their meaning get lost along the way. For example, lines 41-46: “At present, the research…phase change energy storage materials.” Lines 53-56, lines 85-90 ,and much more.

I have modified it.

Figures 1 and 2 can be combined to make a comparison between the ideal and actual phase change material behavior.

I have modified it.

In introduction line 95, Therefore …“hot melt method”… It appears that this method has not been mentioned prior this sentence, and should be elaborated.

I have modified it.

What is “reverse thinking method” in line 112? It would be best if you used the proper terminology.

I have changed it to inverse method

The sentences in lines 41-45 and lines 124-127 are similar, almost redundant. Please, consider revising.

I've already integrated and deleted.

In the “materials and methods section”, what is the composition of your material. Is it equimolar or equi-mass mixture of salt hydrates MgCl2.6H2O -CaCl2.6H2O? The detail on the preparation of the mixture is unclear.

The reagents of the samples were purchased from www.yao123.com. Purity was analytical purity. Taking CaCl2·6H2O-MgCl2·6H2O-2% SrCl2·6H2O-0.3% HEC as the research object, the heat storage and release experiments were carried out by changing the heat storage temperature, cooling temperature, cooling rate, the content of additive in the composite phase change material, ultrasonic and other influencing factors. Phase change materials are independently developed by the laboratory.

Figure 3 caption is not correct? Isn’t step heating process? Also, consider combining Figure 3 and 4 as they portray the same idea: step heating and cooling process. Likewise, combine Figures 5 and 6 of the DSC.

The constant temperature tank shown in Figure 3 has the functions of cooling and heating. But heating and cooling are done in stages, using the same instrument.

Check the legend of Figure 7(a), I think there is a mixup of green and black dash lines. Don’t you think? Check also the caption of Figure 7(a). step cooling?

I have modified it.

Table 2 and Figure 8 present the effects of heat charging temperature on the phase transition hysteresis. Consider using only one to avoid redundancy. Similarly, consider revising throughout the manuscript, Table 3 and Figure 10, and so on.

The table shows the data values and the graph shows the trend.

What is the heating rate in the DSC analysis of Figure 9?

Heating rate is 5°C per minute.

It is very difficult to understand the experimental procedure in section 3.2.1, especially lines 251-255. In fact, the experimental procedure is uncoherent: in section 3.1.2 the PCM is heated in a water bath and then cooled in natural convection, while in section 3.2.1 the PCM is heated in water and cooled in alcohol? Why are you choosing two different methods of cooling ? Those are some concerns that make this study unreliable.

Because in the constant temperature tank, the heating medium is water, and the cooling medium is alcohol. Alcohol is very volatile.Using constant temperature tank heating and cooling can accurately control the target temperature, using air cooling cooling can reflect different cooling rates.

In Table 4, where do you get the melting point values, as the heating step was not mentioned in the experimental section.

The phase transition point of the material is 25℃, and the latent heat of phase transition is173.9 kJ·kg-1.

The cooling and heating rate of the DSC in Figure 14 are not given.

The rate is consistent at 5°C per minute.

Why did you choose different starting weight of materials, 50 mg in section 3.1 and 40 mg in sections 3.2 and 3.3?

We experimented with two different initial weights and there was no effect on the results.

In lines 351-356, it is said that the samples were heated to 50 o what was the heating rate? Was it variable as shown in Figure 18 (a)?

Heating rate is 5°C per minute.

Therefore, sections 3.1.1, 3.2.1 and 3.3.1 should be moved to the experimental sections.

I recognize that specific experimental conditions in the results analysis section can reflect the change of prominent experimental conditions.

Furthermore, the main factors of phase change hysteresis were not given. The authors only discussed the exogenous factors such as heat storage temperature, cooling storage temperature. However, how about the endogenous factors? The authors should elaborate on that or give some hints for future directions .

It is discussed in the results and added in the conclusion.

Round 2

Reviewer 1 Report

The authors improved the manuscript by implementing the required corrections. 

Author Response

Thanks.

Reviewer 2 Report

1.Figure 5 (a) still has a problem. The legend and the graph are not consistent. green and black lines are mixed up.

2. In line 346, is the heating of phase change material leads to a solidification? not a true statement.

3. The explanation in line 367 is also unclear. The heating of the PCM does not lead to a solidification.

Author Response

I've changed everything.